Effect of freshwater mussels on the vertical distribution of anaerobic ammonia oxidizers and other nitrogen-transforming microorganisms in upper Mississippi river sediment

Black Ellen M. 1
Chimenti Michael S. 2
Just Craig L. craig-just@uiowa.edu 1
1 Department of Civil and Environmental Engineering, University of Iowa , Iowa City , IA , United States of America
2 Iowa Institute of Human Genetics, University of Iowa , Iowa City , IA , United States of America
Esteban María Ángeles
Electronic publication date: 2017 Jul 12
Publication date: 2017
Volume: 5
Electronic Location ID: e3536
Received 2017 Jan 4; Accepted 2017 Jun 13
Copyright: ©2017 Black et al.
Copyright year: 2017
Copyright holder: Black et al.
License: This is an open access article distributed under the terms of the Creative Commons Attribution License, which permits unrestricted use, distribution, reproduction and adaptation in any medium and for any purpose provided that it is properly attributed. For attribution, the original author(s), title, publication source (PeerJ) and either DOI or URL of the article must be cited.
License URL: https://creativecommons.org/licenses/by/4.0/

Keywords: Freshwater mussels, Anaerobic ammonia oxidizers, Anammox, Nitrogen cycle, Sediment microbiology

Funding: IIHR - Hydroscience & Engineering and the University of Iowa Water Sustainability Initiative This work was supported by IIHR - Hydroscience & Engineering and the University of Iowa Water Sustainability Initiative. The funders had no role in study design, data collection and analysis, decision to publish, or preparation of the manuscript.

==============================
Targeted qPCR and non-targeted amplicon sequencing of 16S rRNA genes within sediment layers identified the anaerobic ammonium oxidation (anammox) niche and characterized microbial community changes attributable to freshwater mussels. Anammox bacteria were normally distributed (Shapiro-Wilk normality test, W-statistic =0.954, p = 0.773) between 1 and 15 cm depth and were increased by a factor of 2.2 (p < 0.001) at 3 cm below the water-sediment interface when mussels were present. Amplicon sequencing of sediment at depths relevant to mussel burrowing (3 and 5 cm) showed that mussel presence reduced observed species richness (p = 0.005), Chao1 diversity (p = 0.005), and Shannon diversity (p < 0.001), with more pronounced decreases at 5 cm depth. A non-metric, multidimensional scaling model showed that intersample microbial species diversity varied as a function of mussel presence, indicating that sediment below mussels harbored distinct microbial communities. Mussel presence corresponded with a 4-fold decrease in a majority of operational taxonomic units (OTUs) classified in the phyla Gemmatimonadetes, Actinobacteria, Acidobacteria, Plantomycetes, Chloroflexi, Firmicutes, Crenarcheota, and Verrucomicrobia. 38 OTUs in the phylum Nitrospirae were differentially abundant (p < 0.001) with mussels, resulting in an overall increase from 25% to 35%. Nitrogen (N)-cycle OTUs significantly impacted by mussels belonged to anammmox genus Candidatus Brocadia, ammonium oxidizing bacteria family Nitrosomonadaceae, ammonium oxidizing archaea genus Candidatus Nitrososphaera, nitrite oxidizing bacteria in genus Nitrospira, and nitrate- and nitrite-dependent anaerobic methane oxidizing organisms in the archaeal family “ANME-2d” and bacterial phylum “NC10”, respectively. Nitrosomonadaceae (0.9-fold (p < 0.001)) increased with mussels, while NC10 (2.1-fold (p < 0.001)), ANME-2d (1.8-fold (p < 0.001)), and Candidatus Nitrososphaera (1.5-fold (p < 0.001)) decreased with mussels. Co-occurrence of 2-fold increases in Candidatus Brocadia and Nitrospira in shallow sediments suggests that mussels may enhance microbial niches at the interface of oxic–anoxic conditions, presumably through biodeposition and burrowing. Furthermore, it is likely that the niches of Candidatus Nitrososphaera and nitrite- and nitrate-dependent anaerobic methane oxidizers were suppressed by mussel biodeposition and sediment aeration, as these phylotypes require low ammonium concentrations and anoxic conditions, respectively. As far as we know, this is the first study to characterize freshwater mussel impacts on microbial diversity and the vertical distribution of N-cycle microorganisms in upper Mississippi river sediment. These findings advance our understanding of ecosystem services provided by mussels and their impact on aquatic biogeochemical N-cycling.

Introduction

Native freshwater mussels (Order Unionida) are ecosystem engineers that significantly alter benthic habitats through biodeposition of feces and pseudofeces, rich in ammonium (NH4+) and organic carbon (C), into sediment (Thorp et al., 1998; Vaughn & Hakenkamp, 2001; Bril et al., 2014). The estimated mussel filtration capacity in a 480 km, Upper Mississippi River (UMR) segment, as a percentage of river discharge, is up to 1.4% at high flows, up to 4.4% at moderate flows and up to 12.2% during low flows (Newton et al., 2011). The mussels in this river segment collectively filter over 14 billion gallons of water, remove tons of biomass from the overlying water, and deposit tons of reduced C and nitrogen (N) at the water-sediment interface each day (Newell, 2004). The pocketbook mussel (Lampsilis cardium) and threeridge mussel (Amblema plicata) comprise up to 38% and 56% of the mussel biomass in the UMR, respectively (Newton et al., 2011). A habitat near Buffalo, Iowa, in UMR Pool 16, had mean densities of 1.56 L. cardium-m−2 and 7.18 A. plicata-m−2 that correlated with fine sediment diameters (d50 = 0.300 ± 0.121 mm) which were presumably influenced by mussel burrowing (Young, 2006). Mussels live primarily buried in sediment, with their posterior end often flush with the sediment surface (Haag, 2012), or slightly below the surface in soft sediments (Allen & Vaughn, 2009; Allen, 1923; Matteson, 1955). This positions adult freshwater mussels 6–10 cm into the sediment with tendencies toward more shallow burrowing during the spring and summer (Schwalb & Pusch, 2007). Extensive observations in the UMR concluded that A. plicata were often found with portions of their shell above the water-sediment interface, while L. cardium burrow a few cm into the sediment during the summer (Newton, Zigler & Gray, 2015). Additionally, A. plicata often burrowed up to 2.5 cm vertically (Allen & Vaughn, 2009) in response to stressors while L. cardium moved more horizontally when stressed (Newton, Zigler & Gray, 2015). Two common stressors, that happen to be created by the mussels themselves, are low dissolved oxygen (DO) and elevated ammonia (NH3) and NH4+ (Bril et al., 2017; Haag, 2012). We hypothesize that this frequent vertical and horizontal movement by mussels, many times as an indirect and/or direct response to their own waste production, has a significant impact on porewater chemistry and microbiology in UMR sediments.

The evidence for freshwater mussel impacts on aquatic chemistry is compelling, especially for nutrients. A dense mussel population can sequester 2 g C day−1m−2, 200 mg N day −1m−2, and 50 mg phosphorus day−1m−2 from river water into sediment (Strayer, 2014). During the summer months, biodeposition-derived N from mussels was roughly 67% NH4+, 28% amino acids, and 5% urea (Bayne, 1973). Mussel biodeposition accounted for up to 40% of total N demand in freshwaters and up to 74% of N in the food web, but was sometimes dampened (Atkinson, Kelly & Vaughn, 2014) in high nutrient environments (Atkinson, Kelly & Vaughn, 2014). Our previous work showed mussel burrowing and biodeposition, just below the water-sediment interface, increased porewater NH4+, nitrate (NO3−), nitrite (NO2−), and total organic C concentrations by 160%, 38%, 40%, and 26%, respectively (Bril et al., 2017; Bril et al., 2014). However, the experimental design of our previous work limited our ability to assess the effects of mussels on the broad microbial community that was transforming N simultaneously and, quite likely, synergistically.

Figure 1 Freshwater mussels deposit feces and pseudofeces containing nitrogen and carbon at the water-sediment interface (i.e., oxic-anoxic transition).

NH4+ resulting from mussel biodeposits may be oxidized via nitrification and comammox in oxic conditions (yellow arrows), and/or by anammox and n-damo near the oxic-anoxic interface (gray arrows). Oxidized nitrogen species (NO2− and NO3−) may be reduced by dissimilatory nitrate reduction to ammonium (DNRA) and denitrification pathways (blue).

The UMR is an N-rich agro-ecosystem (Hill et al., 2011; Hill et al., 2008; Houser & Richardson, 2010; Ikenberry et al., 2014; Schilling, Wolter & McLellan, 2015) shown to foster high microbial N transformations (Mason et al., 2016; Millar et al., 2015) potentially making the effects of mussels on a variety of N-transforming bacteria and archaea more pronounced than in any other freshwater environment. The first step in transforming biologically active N is nitrification by aerobic ammonium oxidizing bacteria (AOB) (Fig. 1, yellow arrows), such as the genera Nitrosomonas and Nitrosospira in the Nitrosomonadaceae family (Aakra et al., 2001; Burrell, Phalen & Hovanec, 2001; Hayatsu, Tago & Saito, 2008; Prosser, Head & Stein, 2014), and aerobic ammonium oxidizing archaea (AOA) in multiple candidate genera. AOB and AOA are metabolically diverse (Leininger et al., 2006) and serve a functionally important role of catalyzing the rate limiting step of nitrification (Martens-Habbena et al., 2009) in various freshwater niches. For example, Candidatus Nitrososphaera (phylum Thaumarchaetoa), a group of thermophilic AOA (Hatzenpichler et al., 2008), and AOB species in genera Nitrosospira and Nitrosococcus (Koper et al., 2004) can use urea as an alternative source of NH4+ (Spang et al., 2012). AOA often outnumber AOB (Prosser & Nicol, 2008) due to their ability to grow at NH4+ concentrations below 10 nM (Martens-Habbena et al., 2009), compared to 10 µM for some AOB species (Bollmann, Bar-Gilissen & Laanbroek, 2002). In the second step of nitrification, nitrite oxidizing bacteria (NOB), such as Nitrospira (phylum Nitrospirae), and Nitrobacter, Nitrococcus, and Nitrospina (phylum Proteobacteria) (Prosser, Head & Stein, 2014), aerobically oxidize NO2− to NO3− (Fig. 1, yellow arrows).

Nitrospira are the most abundant and diverse group of NOB and dominate numerous habitats, ranging from freshwater sediment to engineered wastewater treatment plants (Daims et al., 2015; Koch et al., 2015; Lucker et al., 2010). Furthermore, NOB species Nitrospira moscoviensis and Nitrospira lenta can derive NH4+ from urea hydrolysis, provide NH4+ to AOB, and subsequently may oxidize NO2− from AOB in a process deemed “reciprocal feeding of nitrifiers” (Daims et al., 2015; Koch et al., 2015). Candidatus Nitrospira inopinata, can also use urea as an alternative NH4+ source (Daims et al., 2015) and has all the genes necessary for complete ammonia oxidation (comammox) to NO3− (Van Kessel et al., 2015) (Fig. 1, yellow curved arrow). Denitrifiers complete the conventional N-cycle by sequentially reducing NO3− to nitric oxide (NO), nitrous oxide (N2O), and nitrogen gas in anoxic and high C environments (Fig. 1, blue arrows) (Hayatsu, Tago & Saito, 2008).

A specialized group of bacteria in the phylum Planctomycetes, anaerobic ammonium oxidizing (anammox) bacteria, oxidize NH4+, utilize NO2− as a terminal electron acceptor, and produce N2 gas (Kartal et al., 2011; Kuenen, 2008) (Fig. 1, gray arrows). Anammox bacteria thrive at the interface of oxic–anoxic conditions due to dependence on NO2− production by AOB or AOA (Thamdrup, 2012). Anammox and NOB compete for NO2− in low substrate environments, and this is especially true for Nitrospira NOB, which share a homologous form of the key enzyme catalyzing NO2− oxidation with anammox (Lucker et al., 2010). In another example, N. moscoviensis can adapt to a range of oxygen concentrations by coupling formate oxidation and NO3− reduction (Koch et al., 2015). Recently, an N-cycling enrichment culture revealed comammox bacteria co-occurring with anammox bacteria in the genus Candidatus Brocadia, presumably enhanced by the ability of commamox organisms to oxidize NH4+ in low oxygen conditions (<3.1 µM) (Van Kessel et al., 2015). Nitrospira species, N. moscoviensis and “Ca. Nitrospira inopinata” in particular, are examples of NOB which harbor a unique ability to assist or compete with anammox for N-substrate in a variety of niches (Lucker et al., 2010).

Shallow sediments also pose a competitive niche for anammox bacteria because of high NH4+ fluxes into oxic sediment and NO2− limitations from denitrification (Thamdrup, 2012; Trimmer & Engstrom, 2011). Another addition to the suite of known N-transformations includes prokaryotic coupling of anaerobic oxidation of methane with denitrification (Raghoebarsing et al., 2006). In nitrite- and nitrate-dependent anaerobic methane (CH4) oxidation (n-damo) (Raghoebarsing et al., 2006; Thamdrup, 2012; Welte et al., 2016), NO3− reduction to NO2− and NO2− reduction to N2 are coupled with CH4 oxidation to CO2 (Fig. 1, gray line and curved arrows). Nitrate-damo biochemical processes have been linked to family “ANME-2d” (Ding et al., 2016; Haroon et al., 2013), while nitrite-damo was discovered for “Candidatus Methylomirabilis oxyfera”(Ettwig et al., 2010) in phylum “NC10” (Ettwig et al., 2009; Luesken et al., 2011; Padilla et al., 2016), and both are widespread in anoxic freshwater sediments (Deutzmann & Schink, 2011; Ding et al., 2016; Ettwig et al., 2009; Hu et al., 2009).

Mollusks have been shown to influence the diversity of microbial communities and abundance of N-transforming microorganisms. For example, metagenomic profiling revealed a marine California mussel (Mytilus californianus) shell provided a niche for N- and C-transforming microorganism populations (Pfister, Meyer & Antonopoulos, 2010), and a restored oyster reef enhanced nitrification and denitrification rates greater than 10-fold (Kellogg et al., 2013). Furthermore, an experimental microcosm study reported enhanced prokaryotic metabolic activity and diversity following a biodeposition rate of 10 g m−2 d−1 of mussel feces and pseudofeces (Pollet et al., 2015). Additionally, clusters of the zebra mussel (Dreissena polymorpha) in a lake increased heterotrophic bacteria density, activity, and diversity (Lohner et al., 2007). Since the impact of native freshwater mussels on prokaryotic diversity and abundance in the UMR is largely unknown, this study utilized targeted and non-targeted sequencing of the 16S rRNA gene to determine how N-transforming microorganisms and microbial community structure differs in sediments with mussels compared to sediments without mussels.

Materials and Methods

UMR sediments were collected within a dense, well-characterized mussel assemblage in the Buffalo Habitat of UMR Pool 16 (Young, 2006) (41.452804, −90.763299), and from a slightly up-river location with no mussels (41.451540, −90.753275) using a 3-inch diameter, hammer-driven, acrylic tube (Batch 1 samples) or a 2-inch diameter, post-driver sediment sampler with a polypropylene liner (Multi-Stage Sediment Sampler, Batch 2 samples; Art’s Manufacturing and Supply, Inc., American Falls, ID, USA). Batch 1 sediment was used to identify the vertical distribution of anammox bacteria below freshwater mussels. For Batch 1, the acrylic tube for each core (n = 3 with-mussels) was penetrated at 1, 3, 5, 7, 11, and 15 cm sediment depths with a 3/8th-inch diameter, ethanol flame-sterilized drill bit to enable sediment collection. In comparison, Batch 2 sediment was used to characterize anammox abundance, microbial diversity, and community structure in shallow sediments below mussels. For Batch 2, the polypropylene liner for each sediment core (n = 5 with-mussels, n = 5 no-mussels) was penetrated at depths of 3 cm and 5 cm. Sediment was sampled for DNA isolation (in quadruplicate) for a combined sample size of n = 20 for 3 cm depth with-mussels, n = 20 for 5 cm depth with-mussels, n = 20 for 3 cm depth without mussels, and n = 20 for 5 cm depth without mussels. Genomic DNA was isolated from 0.25 g of each sediment sample (PowerSoil® DNA Isolation Kit; MoBio Laboratories, Inc., Carlsbad, CA, USA) and stored at −20°C. Batch 2 Genomic DNA was used for anammox-targeted qPCR (n = 20 for each treatment) and 16S rRNA gene amplicon sequencing (n = 10 for each treatment).

Anammox 16S rRNA gene quantification

Microbial culture from a sidestream deammonification process (Hampton Roads Sanitation District, Virginia Beach, VA, USA) served as a source of anammox genetic material for qPCR standard curve construction. PCR products (primers A483f (5′-GTCRGGAGTTADGAAATG-3′) and A684r (5′-ACCAGAAGTTCCACTCTC-3′) (Sonthiphand & Neufeld, 2013)) of the anammox 16S rRNA gene was purified with Qiaquick PCR purification Kit (Qiagen Inc.; Valencia, CA, USA), and cloned into the pCR 2.1-TOPO® vector using the TOPO® TA cloning Kit (Invitrogen Corp.; Carlsbad, CA, USA). Clones were Sanger sequenced at the University of Iowa Institute of Human Genetics with M13F (5′-TGTAAAACGACGGCCAGT-3′) and M13R (5′-CAGGAAACAGCTATGAC-3′) primers to ensure anammox 16S rRNA PCR products were inserted into the vector. Nucleotide sequences were aligned using the Standard Nucleotide Basic Local Alignment Search Tool (Altschul et al., 1997) (GenBank Accession: KU047953) and classified as Candidatus Brocadiales (of the Planctomycetes phylum) with a 95% confidence threshold using RDP Naïve Bayesian rRNA Classifier Version 2.10 (Wang et al., 2007). Plasmid DNA concentration was quantified with Qubit® Fluorometer 1.0 (Thermo Fisher Scientific, Inc.; Waltham, MA, USA), serially diluted, and used to construct qPCR calibration curves.

The anammox 16S rRNA gene from batches 1 and 2 was quantified (Wang et al., 2015) with qPCR using QuantStudio™ 7 Flex Real-Time PCR System (Thermo Fisher Scientific, Inc.; Waltham, MA, USA) with primers A483f and A684r (Sonthiphand & Neufeld, 2013) and analyzed with QuantStudio™ Real-Time PCR Software (Thermo Fisher Scientific, Inc.; Waltham, MA, USA). The threshold cycle (Ct) curves were satisfactory (slope = − 3.374, Y-int = 36.702, R2 = 0.998, and amplification efficiency = 97.99%), and PCR product dissociation curves revealed singe peaks centered at a melting temperature of 83°C. The statistical significance of 16S rRNA gene copies was determined via a one-way, repeated measures analysis of variance (ANOVA) (SigmaPlot 13.0, Systat Software, Inc., Chicago, IL, USA) between the 4 treatment groups (n = 20) following a passed normality test (p = 0.826, Shapiro–Wilk) and an equal variance test (p = 0.073, Brown-Forsythe). Pairwise multiple comparison procedures were completed via the Holm-Sidak method with a significance level of 0.050 and a power of 0.990.

Non-targeted amplicon sequencing of the 16S rRNA gene

Batch 2 genomic DNA (20 µL, 1–50 ng/µL) was analyzed by the Argonne National Laboratory, Environmental Sample Preparation and Sequencing Facility (ESPSF) utilizing the Earth Microbiome Project protocol (http://www.earthmicrobiome.org/emp-standard-protocols/16s/). All samples were analyzed together in one batch. The v4 region of prokaryotic 16S rRNA gene (515F-806R) was amplified using the following conditions: 3 min at 94°C, 35 cycles of 94°C for 45 s, 50°C for 60 s, and 72°C for 90 s, followed by 10 min at 72°C (Caporaso et al., 2012). The PCR mixture consisted of 13.0 µL PCR grade water, 10.0 µL 5 PRIME HotMasterMix (Quanta Biosciences, Beverly, MA, USA), 1.0 µL genomic DNA, and 0.5 µL forward and reverse primers (10 µM). 16S rRNA gene amplicon libraries were sequenced by ESPSF using Illumina MiSeq paired end reads (2 × 151 bp) (Caporaso et al., 2012) and uploaded to MG-RAST (ID’s: 4705672.3–4705709.3) and NCBI (BioProject ID PRJNA374585).

Determining the operational taxonomic units (“OTUs”) in each sample from the raw 16S rRNA gene amplicon reads was accomplished using the default Quantitative Insights into Microbial Ecology (QIIME) open-reference pipeline (Navas-Molina et al., 2013). Briefly, the QIIME open-reference pipeline takes paired-end reads as input, which are then joined, demultiplexed, filtered, and clustered into OTUs with uclust (Edgar, 2010). Representative sequences from each cluster were aligned (Caporaso et al., 2010) to GreenGenes 13.5 reference database (DeSantis et al., 2006) with a 97% similarity threshold. RDP classifier (Wang et al., 2007) was used for taxonomy assignment, PyNAST (Caporaso et al., 2010) was used for multiple sequence alignment. Phylogenetic trees were constructed using FastTree2.1.3 with default settings (Price, Dehal & Arkin, 2010). The OTU table from QIIME open reference picking (‘otu_table_mc2_w_tax_no_pynast_failures_json.biom’ in the standard QIIME workflow) was imported into R using the phyloseq package (McMurdie & Holmes, 2013) for downstream analysis, along with the corresponding phylogenetic tree (‘rep_set.tre’) and a metadata mapping file. These datasets were merged to create a single ‘physeq’ object representing the experiment. Alpha-diversity was calculated on the unfiltered OTU abundance data using the Observed species, Chao1 (Chao & Chiu, 2001), and Shannon (Li et al., 2011) metrics. Beta-diversity was calculated using a matrix of Bray-Curtis (Bray & Curtis, 1957) intersample distances and ordination plots calculated with non-metric multidimensional scaling (NMDS). Differential abundance analysis was carried out using the DESeq2 (Love, Huber & Anders, 2014) R package with default settings (test type was “Wald,” fit type was “parametric”). Translating physeq objects into a compatible DESeq2 object was performed with the “phyloseq_to_deseq2” function. The complete data analysis R script can be downloaded from the public GitHub repository: https://github.com/mchimenti/black_chimenti_just_phyloseq/blob/master/phyloseq.r.

Analysis at the OTU level provided a fine scale resolution for significant differences in microbial ecology between mussel and no mussel treatments. To put these results into a biological context, the genus-level OTU file was used to compare relative abundances for N-cycle phylotypes. These groups include AOA genus Candidatus Nitrososphaera, nitrate-damo family “ANME-2d”, NOB genus Nitrospira, anammox genus Candidatus Brocadia, AOB family Nitrosomonadaceae, and nitrite-damo phylum “NC10”. Relative abundance counts for each N-cycle group was tested for statistical significance between treatments, using metadata groups “3 cm with-mussels” (n = 10), “5 cm with-mussels” (n = 10), “3 cm no-mussels” (n = 10), and “5 cm no-mussels” (n = 10). 1-way ANOVA’s of each N-cycle group was performed using the Kruskal–Wallis test (p < 0.05) with Dunn’s multiple correction test (Padj < 0.05) (GraphPad Prism 7.0; La Jolla, CA, USA). Similarly, multiple comparisons were made between all N-cycle phylotype groups and their respective treatments (n = 10); significant differences between relative abundances were tested using the Kruskal–Wallis test (P < 0.0001) and Dunn’s multiple comparison test (Padj < 0.05).

Results

Anammox-targeted 16S rRNA gene quantification

The targeted 16S rRNA gene data from Batch 1 (n = 3, with-mussels) indicated an anammox bacterial gene copy maximum (∼3 ×105 copies g−1 sediment) between 3 cm and 7 cm sediment depth in the presence of mussels (Fig. 2A). The Batch 1 data was normally distributed between 1 cm and 15 cm (Shapiro–Wilk normality test, W-statistic = 0.954, p = 0.773). Only one sediment core went beyond 7 cm leaving anammox bacterial gene copy data at 11 cm and 15 cm without replicates. The Batch 2 data (n = 20 for 3 cm with-mussels, n = 20 for 5 cm with-mussels, n = 20 for 3 cm no-mussels, n = 20 for 5 cm no-mussels) showed that anammox bacteria experienced a 2.2-fold increase (p < 0.001) at 3 cm with-mussels compared to the no-mussels control (Fig. 2B). The anammox gene copies measured at 5 cm were statistically indistinguishable between the with-mussels and no-mussels treatments.

Figure 2 (A) The mean anammox 16S rRNA gene copies (per gram of sediment) in the presence of mussels were normally distributed (Shapiro–Wilk normality test, W-statistic =0.954, p = 0.773) with depth (Batch 1 data). Error bars represent 1 standard deviation from the mean. (B) Mussels (salmon-colored data) significantly increased the anammox 16S rRNA gene copies at 3 cm depth (p < 0.001; Batch 2 data).

The anammox gene copies were statistically indistinguishable with mussels as compared to the no mussels (turquoise-colored data) sediments at 5 cm (Batch 2 data). The outer most open circles in Fig. 2B represent data outliers, box boundaries represent the 25th and 75th percentile, the line within the box is the median, and error bars indicate 10th and 90th percentiles.

Non-targeted sequencing of the 16S rRNA gene

Summing across all samples, a total of 2,103,661 amplicon sequences were analyzed and about 76,000 unique OTUs were reported by QIIME. Of the unique OTUs, 18,777 had 10 or more reads and 3,916 OTUs had counts exceeding 100 reads. Mussel bed samples had read counts of 45,290 (±15,271) at 3 cm sediment depth and 52,451 (±7,044) at 5 cm sediment depth, while no-mussel samples had 48,920 (±7,517) read counts at 3 cm depth and 63,706 (±25,379) at 5 cm sediment depth (read depths depicted in Fig. S1). The top phyla in mussel bed sediments were Proteobacteria (40.7%), Nitrospirae (35.2%), Chloroflexi (5.9 %), Euryarchaeota (5.0%), Chlorobi (4.2%), and Bacteroidetes (2.3%). Proteobacteria decreased by about 6% with mussels while Nitrospirae increased by 10% with mussels. The most abundant taxonomic families in the Nitrospirae phylum were Thermodesulfovibrionaceae (55%), “FW” (33%), and Nitrospiraceae (13%), and were 5% less, 3% and 2% greater than in no-mussel samples, respectively. With mussels, Proteobacteria taxonomic classes consisted of the following proportions: 68% Deltaproteobacteria (8% less than without-mussels), 16% Gammaproteobacteria, and 15% Betaproteobacteria. A majority of these Deltaproteobacteria OTUs were from “BPC076”, Desulfarculales, and Syntrophobacterales taxanomic orders, while orders Burkholderiales and Xanthomonodales made up a majority of Betaproteobacteria and Gammaproteobacteria taxons.

Species richness was analyzed using three common measures: Observed species, Chao1 and Shannon indices (n = 20 with-mussels and n = 20 without mussels). Together, the three measures indicated a decrease in microbial community richness and evenness in the presence of mussels as compared to sediments without mussels (Fig. 3A). The observed decrease in alpha-diversity reached significance for each of the three measures tested (p = 0.0054 or lower). A similar result was obtained when calculating alpha-diversity measures in samples exclusively from 3 cm (n = 10) or exclusively from 5 cm (n = 10) depths in the presence and absence of mussels. However, the decrease in richness was more pronounced at 5 cm than at 3 cm depth (Figs. S2 and S3).

Figure 3 (A) Sediments with mussels have lower observed species richness (p = 0.005), Chao1 diversity (p = 0.005), and Shannon (p = 0.0003) diversity than no-mussel sediments. (B) NMDS analysis using Bray-Curtis distances revealed sample clustering as a function of mussel presence, but not sediment depth.

To compare intersample diversity in species abundances and community composition (“beta diversity”), we employed NMDS scaling to accurately visualize, in 2D space, the higher-order community structure between with-mussels and no-mussels samples (Fig. 3B). The NMDS model produced an excellent representation of the bray–curtis distances for all samples (convergence in 20 iterations, stress ∼ 0.06; shepard plot shown in Fig. S4). The beta diversity clearly differentiated as a function of mussel presence, but not sediment depth (Fig. 3B). Taken together, these data show that mussel presence had a pronounced influence on the microbial community evenness, richness, and composition within the sediment.

Differential abundances in OTUs did not reach significance for metadata values of sediment depth or comparisons between sediment cores. On the other hand, there were numerous differences in OTU abundances when comparing sediment with mussels and without mussels. We performed a differential abundance estimation with the DESeq2 R package using mussel presence status (n = 20 with-mussels, n = 20 no-mussels) as our covariate. 734 OTUs (or 0.94% of the 77,288 OTUs tested) reached significance with a false discovery rate of 0.01. The vast majority of OTUs belonging to the phyla Gemmatimonadetes, Actinobacteria, Acidobacteria, Plantomycetes, Chloroflexi, Firmicutes, Crenarcheota, and Verrucomicrobia decreased by at least 4-fold in the presence of mussels. In contrast, Proteobacteria showed a marked decrease in order Alphaproteobacteria, while showing mixed increasing and decreasing OTUs among Beta-, Delta-, and Gammaproteobacteria. Phylum Nitrospirae also had 38 OTUs which were differentially abundant with p-adj < 0.001. OTUs assigned to the GreenGenes taxonomic family of “0319-6A21” were the most abundant among those OTUs increasing without mussels, while families Thermodesulfovibrionaceae and “FW” were most abundant among those OTUs increasing with mussels.

Many of the Nitrospirae taxons that increased without mussels did so from a smaller average abundance (17 average counts for Nitrospira and up to 126 average counts for Thermodesulfovibrionaceae) relative to those that were increased with mussels (209 average counts for Nitrospira and up to 581 average counts for Thermodesulfovibrionaceae). This explains the 10% increase in Nitrospirae abundance when summing across all samples with mussels. Figure 4 shows the Log2FC categorized by phyla for OTUs with p-adj < 0.0001 (to enhance visual clarity). Significant differences within the Nitrospirae phylum were represented by increases of genus “HB118” in family Thermodesulfovibrionaceae (2.0Log2FC from a mean count of 52, p < 0.001) and unclassified Nitrospira species (0.8Log2FC from an average count of 209, p < 0.001) with mussels. No-mussel treatments showed increases in genus “LCP-6” from family Thermodesulfovibrionaceae (3.6Log2FC from an average count of 126, p < 0.001) and unclassified Nitrospira species (2.1Log2FC from an average count of 17, p < 0.001).

Figure 4 Results from a DESeq2 differential abundance analysis expressed as Log2FC comparison of with-mussels and no-mussels samples.

Negative Log2FC represent phyla enhanced in the mussel bed and each point represents an individual OTU. To enhance clarity, only those OTUs with p-adj <0.0001 are shown.

Despite seemingly even representation of phylum Thaumarchaeota between treatments, unclassified species from Candidatus Nitrososphaera were enhanced from an average abundance of 126 (1.73Log2FC, p < 0.001) without mussels, and AOA species, Candidatus Nitrososphaera gargensis increased from an average count of 16 (2.85Log2FC, p < 0.001) without mussels. One OTU classified in the anammox genus, Candidatus Brocadia, increased from an average count of 17 (3.72Log2FC, p < 0.001) without mussels, while another OTU classified as an unknown Candidatus Brocadia species increased from a mean count of 16 (1.2Log2FC, p = 0.001) with-mussels. Furthermore, OTUs belonging to the AOB family Nitrosomonadaceae increased from an average abundance of 6 (1.9Log2FC, p < 0.001) with mussels. Without mussels, taxonomic groups capable of nitrite-damo, phylum “NC10”, increased from average abundances up to 130 (4.4 Log2FC, p < 0.001), and nitrate-damo family “ANME-2d” increased from average abundances up to 59 (3.4 Log2FC, p < 0.001). A summary of Log2FC values for OTUs relevant to N-transformations are listed in Table S1.

N-cycle phylotypes were examined for statistically significant relative abundances between treatments of mussel presence and sediment depth (Table 1). Candidatus Nitrososphaera experienced a 2.6-fold decrease (p = 0.047) with mussels at 5 cm sediment depth. ANME-2d was three times greater (p = 0.049) at 5 cm sediment depth without mussels, compared to 3 cm sediment depth without mussels. Within the mussel bed, Nitrospira were 1.7 times greater (p = 0.0497) at 3 cm depth, and experienced a 1.9-fold increase (p = 0.025) with mussels at 3 cm sediment depth versus control. Candidatus Brocadia was three times greater (p = 0.013) at 5 cm depth without mussels versus 3 cm without mussels, and the 3 cm sediment showed a 2-fold increase (p = 0.002) with mussels versus control. Nitrosomonadaceae was 2.7 times greater (p = 0.015) at 3 cm with mussels versus 5 cm depth with mussels.

Table 1 The percent relative abundance of N-cycle organisms for mussel and depth treatments.

Taxonomic classification	N-cycle classification	Mean percent relative abundance	
		3 cm with-mussels	3 cm no-mussels	5 cm with-mussels	5 cm no-mussels	
Candidatus Nitrososphaera	AOA	0.26	0.44	0.22	0.58	
ANME-2D	Nitrate-damo	0.12	0.21	0.11	0.63	
NC10	Nitrite-damo	0.0039	0.02	0.0035	0.08	
Nitrospira	NOB/comammox	1.92	1.00	1.11	0.85	
Candidatus Brocadia	Anammox	0.10	0.05	0.07	0.15	
Nitrosomonadaceae	AOB	0.27	0.13	0.10	0.08	

Relative abundances of N-cycle phylotypes were compared within each treatment (Figs. 5A, 5B, 5D, 5E) and between treatments (Figs. 5C, 5F, 5G–5I). Within 3 cm sediment samples with mussels (Fig. 5A), Nitrospira was statistically greater in abundance than Candidatus Brocadia, and ANME-2d was less abundant than Nitrospira. Sediment without mussels at 3 cm depth (Fig. 5B) contained statistically greater abundances of Candidatus Nitrososphaera than Candidatus Brocadia, and greater Nitrospira abundances compared to Candidatus Brocadia, Nitrosomonadaceae, and ANME-2d.

Relative abundance comparisons between mussel and no-mussel treatments at 3 cm depth (Fig. 5C) showed that Candidatus Nitrososphaera was reduced in the mussel treatment, while Nitrospira and Candidatus Brocadia were enhanced with mussels. Within mussel sediment samples at 5 cm depth, Nitrospira was more abundant than Candidatus Brocadia, ANME-2d, and Nitrosomonadaceae. (Fig. 5D). On the other hand, Candidatus Nitrososphaera and Nitrospira were both more abundant than Nitrosomonadaceae without mussels at 5 cm sediment depth (Fig. 5E). Comparing microbial communities at 5 cm depth between mussel and no-mussel treatments (Fig. 5F) revealed that Candidatus Nitrososphaera was less abundant with mussels versus the no-mussel population. Nitrospira and Nitrosomonadaceae phylotypes were more prominent with mussels in shallow sediment depths (Fig. 5G).

Figure 5 Image of N-cycle phylotype comparisons between treatments.

Statistically significant differences in N-cycle organism abundances (P < 0.05). Statistical significance was determined by non-parametric ANOVA with Dunn’s multiple correction test. All boxes show y-axes compared to the “baseline” x-axes, with no boxes representing comparisons not meeting significance. (A–B), (D–E), Comparisons within treatment conditions of mussel presence and depth. (C) Differentially abundant organisms between 3 cm Mussel and 3 cm No Mussel treatments. (F) Abundance comparisons between 5 cm Mussel and 5 cm No Mussel treatments. (G) Differential N-cycle organism abundance between 3 cm Mussel and 5 cm Mussel samples. (H) Comparisons between 3 cm No Mussel and 5 cm No Mussel treatments. (I) Abundance comparisons of 3 cm Mussel versus 5 cm No Mussel, and 3 No Mussel versus 5 Mussel samples.

Overall, Nitrospira made up larger proportions of microbial communities with and without mussels compared to many N-cycle organisms, especially Candidatus Brocadia and Nitrosomonadaceae (Figs. 5C, 5F, and 5G–5I). Without mussels at 3 cm sediment depth, Candidatus Brocadia made up a smaller proportion of the N-cycling microbial community, especially when compared to Candidatus Nitrososphaera, ANME-2d, NC10, and Nitrospira in deeper sediments (Fig. 5H).

Discussion

Numerous studies have found Proteobacteria to be the most abundant phylum in freshwater sediments (Bucci et al., 2014; Dai et al., 2016; Wakelin, Colloff & Kookana, 2008; Zeng et al., 2008; Zhang et al., 2015), sediments with mollusks (Fernandez et al., 2014; Lee et al., 2015), and also mollusk microbiomes (Frischer et al., 2000; Neta et al., 2015; Ngangbam et al., 2015; Trabal et al., 2012). Although our results showed Proteobacteria were the most abundant phylum, we observed a decrease in Proteobacteria by 6% and an increase in Nitrospirae by 10% in the presence of mussels. Families Thermodesulfovibrionaceae and “FW” accounted for many of the Nitrospirae OTUs that increased with mussels and helps explain decreases in species richness for mussel bed sediment.

Sediments contain the most phylogenetically diverse microbial communities (Lozupone & Knight, 2007) and structure and diversity of soil microbial communities is often determined by soil biogeochemistry (Fierer & Jackson, 2006), further supporting the impact mussels have on biogeochemical cycling. In support of our hypothesis, our data indicated that mussel presence in the UMR had a pronounced influence on the microbial community evenness, richness, and composition within the sediment. The observed changes in sediment microbial community structure and diversity showed mussels created a niche for specific microorganisms and may be attributable to the diverse chemical composition of mussel biodeposits, mixing of sediment from mussel burrowing, or the microbes living on mussels. Our findings of distinct microbial communities in mussel bed sediment are corroborated by a study of the California mussel (Pfister, Gilbert & Gibbons, 2014) where taxonomic richness increased and taxa evenness increased following the removal of mussels from a rocky shore habitat.

In contrast to our results of decreased microbial diversity with freshwater mussels, research has shown invasive zebra mussels (Dreissena polymorpha) increased bacterial community diversity and richness (Lee et al., 2015), and metabolic diversity and activity in freshwater sediments (Lohner et al., 2007). Increased microbial diversity and activity has been attributed to the variety of C and N components in feces and pseudofeces, and also selects for the dominant microbial species (Lohner et al., 2007; Pollet et al., 2015). An experiment combining estuarine bivalve species (N. virens, M. arenaria, and M. balthica) implicated mussel-induced changes in O2, NH4+ and NO3− fluxes for the alteration of microbial community composition (Michaud et al., 2009). On the other hand, investigation of microbiota in Thick-shelled River Mussel (Unio crassus) beds did not find any difference in microorganism diversity, abundance, and composition (Richter et al., 2016). This may be explained by the drastic differences in the study site, with high mussel densities (23–433 mussels/m2) and control plots containing low microbial diversities with mean species richness of 48 OTUs/sample with high evenness (Richter et al., 2016). The contrasting findings of microbial community diversity and composition indicate that mussel density and/or mollusk species may produce different responses by microorganism communities.

Additionally, alterations in sediment microbial community structure may arise from exposure to the mussel shell, tissues, or fecal microbiome. Mussel tissue and fecal material has been shown to contain less diverse microbiomes than the surrounding water and sediment for the zebra mussel (Frischer et al., 2000), tropical oyster (Crassostrea rhizophorae) (Neta et al., 2015), and marine mussel, Mytilus californianus (Frischer et al., 2000; Pfister, Gilbert & Gibbons, 2014). Some studies have attributed immediate increased sediment microbial activity to the mussel intestinal microbiome (Grenz et al., 1990). Furthermore, mollusk biodeposition rates and biodeposit chemical compositions are highly dependent on mollusk species (Hegaret, Wikfors & Shumway, 2007; Tenore & Dunstan, 1973), and food availability (Bril et al., 2017; Cranford et al., 2007; Vaughn & Hakenkamp, 2001), so it makes sense that our results differ from studies with dissimilar mollusk species, densities, and study location.

Changes in mussel bed sediment microbial communities was also likely enhanced by mussel burrowing, because diffusion of substrates across the water-sediment interface is a relatively slow process (Kristensen et al., 2012) and is increased by mollusk burrowing (Vaughn & Hakenkamp, 2001), which ultimately affects microbial communities. For example, the burrow of shrimp species Upogebia deltaura and Callianassa subterranean contained distinct bacterial communities and a 3-fold increase in taxon richness (Laverock et al., 2010), and the estuarine bivalve, C. fluminea, stimulated microbial diversity via bioturbation (Novais et al., 2016). It is likely that UMR mussel bed sediments also experience the benefits from bioturbation, such as sediment mixing (McCall, Tevesz & Schwelgien, 1979) and aeration (Vaughn & Hakenkamp, 2001). Furthermore, bioturbation has been linked to increased NH4+ concentrations which alters the N-transforming microbial community (Chen & Gu, 2017), with greatest effects on bacteria growth found at 4–6 cm depth below the water-sediment interface (McCall, Matisoff & Tevesz, 1986).

N-cycle microbial community

Our research revealed an increase in anammox bacteria abundance 3 cm below the water-sediment interface when mussels were present, shown for the anammox community using anammox-targeted qPCR (2.2-fold increase) and for Candidatus Brocadia using non-targeted 16S rRNA gene amplicon sequencing (2-fold increase). The significance of agreement between these techniques is finding that increases in the genus Candidatus Brocadia are representative for the anammox phylotype as a whole. Candidatus Brocadia may also make up a majority of the anammox community in UMR sediment, as amplicon sequencing did not detect anammox bacteria belonging to other genera. We are confident in these conclusions, as Candidatus Brocadia is often the dominant anammox genus in freshwater sediments (Humbert et al., 2009; Shen et al., 2016; Sonthiphand, Hall & Neufeld, 2014). One study showed that feeding of NH4+, NO2−, NO3−, and acetate led to an 80% enrichment of Candidatus ‘Brocadia fulgida’, signifying that B. fulgida could outcompete anammox species in genera Candidatus Anammoxoglobus and Candidatus Kuenenia, species Candidatus ‘Brocadia anammoxidans’, and even denitrifiers when acetate is present (Kartal et al., 2008). This indicates that Candidatus Brocadia has a distinct ecological niche and can utilize intermediates from anaerobic degradation of organic C to reduce NO3− (Kartal et al., 2008). Therefore, it is possible that a portion of our observed increases in Candidatus Brocadia with mussels was attributable to C biodeposition in the UMR.

Our research also revealed a vertical distribution of anammox bacteria with higher abundances near the sediment surface, which reflects the vertical distribution found in an agricultural field (Shen et al., 2017), oxygen minimum zone (Galán et al., 2009), flooded paddy fields (Shen et al., 2017; Zhu et al., 2011), and an urban wetland (Shen et al., 2015). A vertical anammox distribution has been shown to coincide with NH4+ presence and NO2− production (Oshiki, Satoh & Okabe, 2016; Shen, Xu & He, 2014; Shen et al., 2015; Sun et al., 2014) and anammox “hotspots” occur in zones of low, but not entirely absent, O2 availability (Zhu et al., 2013). Anammox abundance in freshwater sediment can range between 7 × 104 and 8 × 106 gene copies g−1 sediment (Shen et al., 2016), or between 106 and 107 gene copies g−1 sediment in peak NO2− microniches at the oxic–anoxic interface (Nie et al., 2015; Shen et al., 2015; Zheng et al., 2016). Studies have shown anammox bacteria increase 1.5 to 2-fold within their niche (Nie et al., 2015; Zheng et al., 2016), similar to our findings of a 2.2-fold increase in anammox bacteria 3 cm below the water-sediment interfacewith mussels.

Co-occurrence of aerobic NH4+ oxidation and anammox niches are likely due to linked NO2− oxidation and reduction, respectively (Shen, Xu & He, 2014). Interestingly, we found that mussels also enhanced taxa from the AOB family Nitrosomonadaceae and the OTUs made up a greater proportion of mussel bed sediment populations near the water-sediment interface. To this point, the pacific oyster (C. gigas) was found to increase porewater NH4+ and elevate the concentration of NH4+ oxidizing microorganisms (Green, Boots & Crowe, 2012). Furthermore, our previous research (Bril et al., 2014; Bril et al., 2017) showed elevated NH4+ and NO2− in porewater of a similar depth below mussels. It makes sense that these groups of N-transforming bacteria co-occur where their substrate microniches overlap, and is likely enhanced by mussels periodically aerating the sediment (Chen & Gu, 2017). Intermittent aeration has shown to enrich microbial cultures in AOB and anammox bacteria in engineered partial nitritation-anammox processes (Shannon et al., 2015; Yang et al., 2015), and similar to our findings, enriches the anammox genus Candidatus Brocadia (Shannon et al., 2015).

On the other hand, we saw a decrease in Candidatus Nitrososphaera (AOA) with mussels at 3 cm (1.7-fold) and 5 cm (2.6-fold) sediment depths. It makes sense that mussels suppress abundance of AOA since these organisms typically dominate sediment niches with low NH4+ concentrations (Hatzenpichler, 2012; Martens-Habbena et al., 2009). Furthermore, a group of OTUs suppressed by mussels were classified at the species level as Candidatus ‘Nitrososphaera gargensis’, which are partially inhibited by NH4+ concentrations (3.08 mM) much lower than AOB (Hatzenpichler, 2012; Hatzenpichler et al., 2008; Nakagawa & Takahashi, 2015; Pester, Schleper & Wagner, 2011). Furthermore, nitrifier niche partitioning studies using agricultural soil showed that AOB increased in abundance and activity following the addition of urine-derived N, while AOA remained unchanged (Di et al., 2009; Hatzenpichler, 2012; Jia & Conrad, 2009). Therefore, it is possible that mussel biodeposits and an increased flux of agriculturally-fed water into sediment by mussel burrowing enhanced porewater NH4+ composition such that Nitrosomonadaceae out competed Candidatus Nitrososphaera. Our results agree with Chen & Gu (2017), who found bioturbated sediment corresponded with a greater diversity of AOB and lower diversity of AOA microbial communities (Chen & Gu, 2017). On the other hand, our results of decreased abundance of Candidatus Nitrososphaera co-occurring with an increase in Nitrospira is in contrast to previous findings that these organisms may exhibit similar niche partitioning (Pester, Schleper & Wagner, 2011). For example, some species in Candidatus Nitrososphaera can adjust their metabolism for low oxygen availability (Zhalnina et al., 2014) and Nitrospira species are adapted to low oxygen concentrations (Maixner et al., 2006; Nowka, Daims & Spieck, 2015; Zhalnina et al., 2014) and microoxic environments (Schramm et al., 2000). Alternatively, our detected increased abundance of Nitrospira may include species with a variety of environmental niches.

Some Nitrospira species have shown to occupy a niche at oxic–anoxic interfaces, in opposition to NOB with higher O2 tolerances such as those in genus Nitrobacter (Schramm et al., 2000). This supports our mussel-attributed increases in relative Nitrospira abundances (1.9-fold) at 3 cm sediment depths. Although we saw two different Nitrospira OTUs suppressed and enhanced by mussels, the mussel-enhanced OTUs had a larger mean abundance by about 12%. Different NOB OTUs enhanced with and without mussels further suggests that mussel bed sediments harbor specific NOB strains sensitive to microoxic niches. Despite OTU variability, we can conclude that mussels enhance the Nitrospira phylotype, especially near the water-sediment interface where Nitrospira were 1.7-times greater than deeper mussel bed depths.

On the other hand, we did not expect to see an increase in both NOB and anammox phylotypes due to competition of NO2− as a substrate. The co-occurrence of Nitrospira and anammox bacteria may be explained by the metabolic versatility of Nitrospira species, especially if mussel-derived urea provided an additional source of NH3 and NO2− via reciprocal feeding between ammonia oxidizers and Nitrospira. Furthermore, these phylotypes have been shown to coexist in an oxygen minimum zone, where anammox bacteria obtained a majority of NO2− from NO3− reducers (Lam et al., 2009). The similar effect size of mussels on Nitrospira (1.9-fold) and Candidatus Brocadia (2-fold) at 3 cm depth suggests that mussels may exert similar influences on the niches of these phylotypes. It is possible that these anammox and Nitrospira phylotypes were functionally linked in shallow mussel bed sediment, which has been shown for microoxic niches (Van Kessel et al., 2015). Furthermore, it is possible that the Nitrospira co-occurring with Candidatus Brocadia were Nitrospira species with the genetic potential for comammox, as a fluorescence in-situ hybridization study confirmed the extensive aggregation of the 2 phylotypes in hypoxic conditions (<3.1 µM O2) (Van Kessel et al., 2015). Despite Nitrospira comammox being identified in numerous aquatic environments (Chao et al., 2016; Daims et al., 2015; Pinto et al., 2016), we cannot conclusively identify comammox without sequencing the ammonia monooxygenase gene (Pinto et al., 2016; Van Kessel et al., 2015).

In contrast to studies which found significant N-reduction on both a marine mussel (Mytilus californianus) (Pfister, Meyer & Antonopoulos, 2010) and a freshwater mussel (Limnoperna fortunei) (Zhang, Cui & Huang, 2014), our results showed that mussels suppressed n-damo OTUs in phylum “NC10” (2.1-fold) and family “ANME-2d” (1.8-fold). One study determined NO3−-damo was responsible for NO3− reduction and anammox for NO2− reductions in a bioreactor supplied with NH4+, NO2−, NO3−, CH4, and anoxic conditions, thus concluding anammox outcompeted NO2−-damo (Hu et al., 2015). These findings make sense, because anammox bacteria have a higher affinity for NO2− (Luesken et al., 2011), anammox outperform n-damo in bioturbated sediments with higher NH4+ and lower NO2− and NO3− (Chen & Gu, 2017), and anammox and n-damo communities have a competitive relationship in burrowed mangrove sediment (Chen & Gu, 2017). Furthermore, NC10 bacteria in a peatland were most prevalent at depths with porewater CH4 concentrations near 300 µM, where NO3− consumption exceeds production, and in completely anoxic conditions (Zhu et al., 2012). According to the literature, it makes sense that we found UMR mussels enhanced Candidatus Brocadia and suppressed NO2− reducing-NC10. Perhaps n-damo organisms did not have a favorable niche in mussel bed sediment because biodeposition products created an excess of NH4+ in sediment porewater (Winkler et al., 2015), or burrowing activity increased oxygen concentrations and made methane oxidation unfavorable (Van Bodegom et al., 2001). Our finding that no-mussel sediment contained three times more ANME-2d in deeper, and presumably anoxic sediment, further suggests that mussels broaden the oxic–anoxic interface niche (Chen & Gu, 2017; Luesken et al., 2011). However, we cannot extrapolate these findings to all denitrifying organisms, since denitrifying species are sporadically distributed among various taxonomic lineages, and are difficult to identify solely with16S rRNA amplicon sequencing (Ishii et al., 2011).

Although we observed greater relative abundances of Nitrospira than Candidatus Brocadia in a majority of treatments, both phylotypes increased by a factor of 2 with mussels at 3 cm depth. No-mussel samples contained a significantly smaller proportion of Candidatus Brocadia in shallow sediments compared to almost all N-transformers found in the deeper control sediments. Our phylotype-level analyses revealed similarities with the OTU-level differential abundance comparisons. For example, phylotype comparisons showed ANME-2d was less abundant than Nitrospira in 3 cm sediments with mussels, and Candidatus Nitrososphaera was more abundant than Candidatus Brocadia in 3 cm sediment samples without mussels. These results relate to DESeq2 OTU comparisons which found Candidatus Brocadia and Nitrospira enhanced with mussels while ANME-2d and Candidatus Nitrososphaera were suppressed with mussels.

Extending our focus beyond N-cycling organisms, we demonstrated that mussels promoted a large effect size for OTUs classified as Thermodesulfovibrionaceae (Nitrospirales order). In contrast to Nitrospira, the Nitrospirales genus Thermodesulfovibrio contains multiple sulfate reducing species (Kirchman, 2012; Sekiguchi et al., 2008) and can outcompete other anaerobic organisms when sulfate is present (He et al., 2015). These findings are corroborated by discoveries of significantly greater C and sulfate concentrations from mussel biodeposits and 63% greater sulfate reduction in sediments with mussels (McKindsey et al., 2011). Biodeposition products often lead to increasingly anoxic sediment and greater activity of anoxic microorganisms (Kellogg et al., 2013; McKindsey et al., 2011), presumably due to consumption of excretion products by oxygen-consuming microorganisms (Pollet et al., 2015). Interestingly, Fdz-Polanco et al. (2001) observed simultaneous N and sulfate removal in an anaerobic fluidized-bed reactor and proposed simultaneous anammox and sulfate reduction. Coupled biological sulfate reduction and anammox reactions are metabolically feasible (Schrum et al., 2009; Strous et al., 2002) and have been of interest in the recent history (Ali et al., 2013; Cai, Jiang & Zheng, 2010; Rikmann et al., 2016; Rios-Del Toro & Cervantes, 2016), therefore warranting further research. Therefore, we showed that Thermodesulfovibrionaceae are significantly increased in the presence of mussels which may affect sulfate reduction (Mahmoudi et al., 2015) in tandem with anammox reactions in UMR sediments.

As a whole, mussels do have an impact on microbial niches and lower the overall community diversity. Mussel-influenced changes in microbiological diversity may have larger ecosystem implications, such as macrobiota richness and diversity (Arribas et al., 2014; Borthagaray & Carranza, 2007). Native freshwater mussels are capable of increasing macrobiota diversity as a result of being keystone species (Hartmann et al., 2016) and ecosystem engineers (Chowdhury, Zieritz & Aldridge, 2016; Lopes-Lima et al., 2014). Mussel biogeochemical hotspots can lead to a bottom-up trophic cascade by enhancing N substrates normally limiting primary productivity, ultimately leading to increased richness (Atkinson et al., 2013) and biodiversity (Allen et al., 2012) of higher trophic levels.

Conclusion

As far as we know, this is the first study to characterize freshwater mussel effects on microbial community diversity, composition, and the vertical distribution of N-cycle microorganisms in the UMR. qPCR of the anammox-specific 16S rRNA gene revealed an increase in anammox bacteria abundance 3 cm below the water-sediment interface when mussels were present, and confirmed anammox bacteria were normally distributed with depth. Non-targeted 16S rRNA gene amplicon sequencing revealed mussel presence suppressed AOA (Candidatus Nitrososphaera) and that the families Thermodesulfovibrionaceae and “FW” (Nitrospirales order) were overrepresented among the enhanced OTUs with-mussels. Mussel bed sediment contained microbial communities with 10% greater Nitrospirae and 6% fewer OTUs belonging to the phylum Proteobacteria, which ultimately had a pronounced influence on microbial community evenness, richness, and composition. This was indicated by lower observed species richness, Chao1 diversity, Shannon diversity, and clustering of mussel samples in an NMDS analysis. We have shown that native freshwater mussels affect niche differentiation of N-cycle microorganisms, as evidenced by increased abundances of AOB family Nitrosomonadaceae, anammox genus Candidatus Brocadia, and NOB genus Nitrospira, while exhibiting a decrease in AOA genus Candidatus Nitrososphaera, and n-damo organisms in the phylum NC10 and family ANME-2d. Co-occurring 2-fold increases in Candidatus Brocadia and Nitrospira in shallow sediment suggests that mussels may enhance microbial niches at the interface of oxic–anoxic conditions, presumably through biodeposition and burrowing. Ultimately, this study demonstrates the large impact mussels have on biogeochemical N-cycling and ecosystem services in freshwater agroecosystems.

Supplemental Information

Supplemental Information 1 Supplemental information

Click here for additional data file.

Additional Information and Declarations

Competing Interests

Author Contributions

DNA Deposition

Data Availability

The authors declare there are no competing interests.

Ellen M. Black and Michael S. Chimenti conceived and designed the experiments, performed the experiments, analyzed the data, contributed reagents/materials/analysis tools, wrote the paper, prepared figures and/or tables, reviewed drafts of the paper.

Craig L. Just conceived and designed the experiments, analyzed the data, contributed reagents/materials/analysis tools, wrote the paper, prepared figures and/or tables, reviewed drafts of the paper.

The following information was supplied regarding the deposition of DNA sequences:

MG-RAST (ID: 4705672.3-4705709.3).

Bioproject Accession: PRJNA374585.

The following information was supplied regarding data availability:

GitHub: https://github.com/mchimenti/black_chimenti_just_phyloseq/blob/master/phyloseq.r.

Project name: Mussel1_longlistminus2.

Static link: http://metagenomics.anl.gov/linkin.cgi?project=mgp18682.

Metagenomes: 4705672.3 through 4705709.3.

Project name: Mussel1_shortlist.

Static link: http://metagenomics.anl.gov/linkin.cgi?project=mgp18674.

Metagenomes: mgm4705417.3; mgm4705418.3.

Individual IDs:

Mussel IDs: 4705672.3, 4705673.3, 4705675.3, 4705676.3, 4705677.3, 4705680.3, 4705681.3, 4705683.3, 4705684.3, 4705690.3, 4705691.3, 4705693.3, 4705695.3, 4705696.3, 4705699.3, 4705705.3, 4705706.3, 4705708.3, 4705709.3, 4705417.3.

No-mussel IDs: 4705674.3, 4705678.3, 4705679.3, 4705682.3, 4705685.3, 4705686.3, 4705687.3, 4705688.3, 4705689.3, 4705692.3, 4705694.3, 4705697.3, 4705698.3, 4705700.3, 4705701.3, 4705702.3, 4705703.3, 4705704.3, 4705707.3, 4705418.3.

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
