# Peer review of "Effect of freshwater mussels on the vertical distribution of anaerobic ammonia oxidizers and other nitrogen-transforming microorganisms in upper Mississippi river sediment"

_PeerJ, doi:10.7717/peerj.3536_

## Round 0.1 · original submission · Major Revisions

The manuscript should be revised following the reviewers' suggestions.

Reviewer 1 ·

Basic reporting

OVERALL REVIEW
The authors use qPCR and amplicon sequencing of 16S rRNA in sediments surrounding freshwater mussels to test for the presence of microbes that use the Anammox metabolism and for the overall patterns of microbial diversity. They find that mussel presence increases anammox gene copies, while lowering overall diversity. The abundance of Nitrospirae were also increased, implying increased nitrite oxidization. We know little about animal effects on microbial diversity and function, so I found this study interesting and think it will be a contribution to the literature. There were also some aspects that need clarification, and I outline those below.

1. There needs to be more detail initially on the species of mussels, their life history, where they live in the sediment (depth, etc), abundance, and any sediment features. These are not described sufficiently to allow any future comparison among studies. We are told that this is a ‘well-characterized mussel assemblage’. !, when there was no characterization, no biological or chemical description and no references. As is, it appears that someone with sequencing expertise went out with a core – this must be remedied prior to publication. The work has little worth without a better biological context.
2. I found some of the discussion of taxa changes confusing. I gather that anammox gene copy was increased without mussels, but the data from v4 16s with Candidatus did not support a change in OTUs. Correct? Probably worth a bit more comment.
3. It is also worth commenting on how Nitrospirae are increasing (nitrite oxidation) without an increase in AOA. Is this due to OUT abundance being a poor predictor of function?
4. Explain replication a bit better. I was confused by the mention of ‘batch’ and wondered if the points in Fig 2 are the replicates.
5. The study should better recognize that much of the function that they are looking for may reside on the mussels themselves and not be free-living in the sediment.

Other edits:
6. p values should be expressed as p<0.001
7. Lots of information was missing from the references cited and this should be corrected.
8. L266 should read ‘…OTU richness increased and taxa equitability increased following the removal . . .” the word ‘rebounded’ seems inappropriate here
9. L83 – what is UMR?

Experimental design

As mentioned above, better describe the replication and df for statistical comparison.

Validity of the findings

As mentioned previously, I found some of the discussion of taxa changes confusing. I gather that anammox gene copy was increased without mussels, but the data from v4 16s with Candidatus did not support a change in OTUs. Correct? Probably worth a bit more comment. It is also worth commenting on how Nitrospirae are increasing (nitrite oxidation) without an increase in AOA. Is this due to OUT abundance being a poor predictor of function?

Must deal with the general information on the system requested above.

Additional comments

The overall review above outlines this.

·

Basic reporting

- very good language
- literature on mussels and their influence on the sediment microbiota is good, but general literature on for instance N-cycle microbes is absent
- Figure S1 is not referenced in the manuscript
- Table S1 requires some reformatting and should contain relative abundance data
- amplicon sequencing data was uploaded to MG-RAST but not NCBI.

Experimental design

- I general valid, although it is a pity that the qPCR data for depths larger 7 cm were not replicated.

Validity of the findings

- I don't completely agree with all of the conclusions and interpretations presented here as I think that some claims are unsupported by the data, while other points need to be clarified and presented better. See below for details.

Additional comments

- The manuscript promises in its title to report changes in anammox and other N-cycle bacteria. However, the authors then do report on changes in anammox abundance, but otherwise only on ammonia-oxidizing archaea and members of the Nitrospirae phylum, most of which however are not nitrite oxidizers as they do not belong to the genus Nitrospira itself. Rather, the OTUs detected appear to mainly be sulfate reducers and not involved in N cycling. Furthermore, the authors present data on the decrease of AOA in the presence of mussels, but fail to present any alternative ammonia-oxidizing microorganisms. Thus, from the data presented it remains completely unclear where anammox would get the nitrite from, and this enigma is also not discussed. I do see indications for betaproteobacterial AOB in the data, and also one genus Nitrospira OTU increases in the presence of mussels and presence of the novel "comammox" Nitrospira (Daims et al. 2015, van Kessel et al 2015) could be an explanation. Furthermore, the authors do mention at one point the possible change in oxygen availability do to increased input of organic carbon, but fail to link this to the observed changes in the nitrifier community.
- In general I miss the promised part on N-cycle bacteria. It remains unclear where the data on nitrate reducers/denitrifying organisms comes from and I see no prove for the claims made on this group in the data. As mentioned above, the discussion of nitrifying organisms is incomplete and inconclusive. The data on anammox is good, but to a certain extend contradictory between the two methods (qPCR vs. amplicon sequencing) used, which also is not discussed/resolved in detail. Further, I do see anaerobic nitrate and nitrite-dependent methane-oxidizing archaea and bacteria (ANME 2d and NC10) decrease in the presence of mussels, which also is not mentioned once. Thus right now the manuscript does not deliver the promises made by the title.
- As also mentioned above, how where denitrifying organisms classified? I don't think this is reliably possible based on 16S rRNA data only. To a certain extent this might be extrapolated from phylogenetic data, but I did not find any data supporting the claim that this guild did not react to the presence of mussels.
- I find the analysis of the amplicon data very incomplete. There is some presentation of the increase or decrease of some apparently randomly selected families, but without any context on why these were selected or what this might imply. As I see the data included in the manuscript I would strongly advise the authors to shift the focus from the (absent) restriction to N-cycle organisms, but rather try to present general changes in all nutrient cycles, as far as possible. Also, a proper presentation of abundance data and shifts of the main OTU/phyla/families is missing right now. Log2fold changes are good for statistical significance, but might be be of minor importance for low-abundance OTUs.
- I would also advise the authors to analyze the amplicon data for differences between the 3 and 5 cm samples in more detail to also see if the effect of mussels on the the main OTUs (and not only the community structure at large) is the same or if there is a depth-dependency.
- One of the main and most surprising findings actually is the reduction in diversity in the presence of mussels. Thus, this finding should be more central in the manuscript as here the data is convincing, unlike for most of the N-cycle organisms.

- It is not entirely clear from the methods and/or results section which data is derived from qPCR specific for anammox, and which from amplicon sequencing. Please indicate clearly in the methods section what the batch 1 and 2 samples were used for, and in the results section if the targeted 16S rRNA data for anammox from the batch 2 data are derived from qPCR or amplicon sequencing.
- Nitrososphaera belong to the Thaumarchaeota phylum, not the Crenarchaeota. The class is Nitrososphaeria, not Thaumarchaeota as indicated in Table S1.
- The presentation of log2-fold changes in Figure 5 and Table S1 is good, but only somewhat meaningful without abundance data (as is shortly mentioned for Nitrospira in lines 204-5). Please include this data in Table S1 and, if possible, in Figure 5. Alternatively, an additional figure showing abundances for all phyla and/or main OTUs in the different sample cohorts would be helpful for the interpretation of the data.


- lines 47-8: "sequester ... from river sediment"? Shouldn't this be "into the sediment" or "filter from the river water"?
- line 50: As the mussels excrete ammonia they can't directly increase nitrate and nitrite.
- lines 51-2: I right now don't see how the ammonia toxicity is connected to this manuscript. Please embed this better, or remove it.
- line 56: I wouldn't call anammox "novel" any more, rather "specialized".
- lines 58-63: How does the manuscript benefit from this historic review of anammox? I would rather recommend spend more time on describing the anammox pathway, as it right now is very unclear where the nitrate comes from (from nitrite oxidation, not from ammonia oxidation!) and why it happens (extra electrons for carbon fixation).
- line 62: anammox doesn't depend on nitrate.
- line 66: As far as I know this estimate is only valid for N losses from marine systems. I don't think any estimates exist on the global scale.
- lines 68-73: The line of argumentation presented here is rather hard to follow and partly misleading. The excretion of ammonia alone can't benefit anammox, as it requires an aerobic ammonia-oxidizing partner to provide nitrite. This can't be the mussel itself, as it reads right now in the manuscript. Furthermore, it needs to be stressed more strongly that anammox requires anoxic conditions, and the link to the excreted organic carbon is unclear. Also, how it reads right now I wouldn't see why nitrite shouldn't be removed by denitrification, which might outcompete anammox when organic C is present under anoxic conditions.
- lines 84-5: The manuscript discusses AOA and putative sulfate reducers in detail. Denitrifying microorganisms are mentioned, but no convincing data is presented. Contrastingly, the untargeted amplicon sequencing approach allows to investigate changes in many taxonomic groups, even if a link to function might be difficult. I thus would say that "and other N-cycle MOs" is wrong.

- lines 106-108: How can whole clone libraries be used for reliable qPCR calibration curves?
- line 133: change to "2 x 151 bp".

- line 161: In figure 2A I see something like 2 or 3x10^5 anammox 16S rRNA gene copies, not 5 as stated here. What is the exact unit here? Per gram sediment?
- line 164: It is quite a pity that the data for 11 and 15 cm is not replicated. Judging by the size of the error bars at 3, 5, and 7 cm there is quite some variation in the data and the observed decrease at 11 (and 15) cm might be due to insufficient sampling only.
- line 188: To lump all Proteobacteria in one group is taxonomically consistent, but rather meaningless as the physiological diversity in this phylum is so large. Please at least split the Proteobacteria into classes (which still is diverse enough).
- line 190: Why "however"?
- line 192: Figure 4 labels them as "Nitrospira".
- lines 195-7: This claim is only partly reflected by the data presented in figure 4. Here at least Gemmatimonadetes and Actinobacteria (on the phylum level) appear to increase in the presence of mussels, as do Chlorobi and Nitrospirae. The resolution on OTU level of course is different, but these contrasting trends are conspicuous and should be discussed.
- line 198-9: I would recommend to present this data for the proteobacterial classes also in all figures.
- lines 208-29: This paragraph reads like a list of hand-selected families arbitrarily picked from the amplicon sequencing data. For most of the families there is no indication why they are mentioned and there is no further link to the manuscript as also no functional correlation is (can be) done. Also, the listing jumps back and forth between families increased and decreased in the presence of mussels without an apparent order. I would strongly advise to restructure this paragraph and include information on how and why the families mentioned here were selected.
- lines 222-3: Contrasting to the text here Figure 5 only indicates one verrucomicrobial OTU to have differential abundance between the different conditions. Please try to have more consistency between main text, figures and tables.
- line 224: Table S1 lists 5 OTUs of Nitrososphaera that increase in the absence of mussels. Why only mention one here? Is this the most abundant OTU?
- line 225: According to Figure 5 there is a decrease of the main crenarchaeal (thaumarchaeal) OTUs, unlike stated here.
- line 229: Since the manuscript wants to discuss N-cycling organisms I find it surprising that this here is the only mention of a putatively nitrite-oxidizing Nitrospira. Even the abstract ignores this and rather mentions a member of the "FW" group of unknown function, which contrastingly is not mentioned here. Further, there is a second genus Nitrospira OTU that decreases 2.05-fold, which is not mentioned here. What does this mean for the nitrite-oxidation potential?
- lines 230-3: This observed difference for anammox between the two data sets needs more attention. Firstly, how do the two Brocadia OTUs differ at the different depths analyzed? is there a shift from 3 to 5 cm that could explain the apparent discrepancy with the qPCR data? Secondly, what are the relative abundances of these OTUs? Do they explain this? Why are you sure the complementary qPCR is more reliable? Did you check if both OTUs detected in the amplicon dataset are targeted by your PCR primers, and that all Brocadia in your clone library are amplified with your general 16S rRNA primers used for the amplicon study?

- line 242-3: per gram sediment?
- lines 245-6: This observed increase of anammox is only true for the 3 cm sample. At 5 cm there even is a slight (but insignificant) decrease. Thus, at least at the limited amount of depths analyzed there could also be a upward shift of anammox activity from 5 to 3 cm, rather then an overall increase.
- line 249: I am not aware of any research reporting the outcompetition of AOB by AOA under high ammonium concentration, but rather the contrary. For this reason I am not so much surprised to see the drop in AOA abundances in the presence of mussels, but I rather wonder about the absence of (reporting of) betaproteobacterial AOB. There is an OTU increasing 1.88-fold that is classified as Nitrosomonadaceae family, this should be analyzed and discussed in more detail.
- line 254: I don't see any data that could support the claim made here about the nitrate reducing bacteria.
- lines 255-60: At the phylum-level resolution presented here I can't think of any valid analyses supporting any of the points discussed here.
- lines 258-60: But at the same time other OTUs increased, so what exactly is the validity of this statement?
- line2 284-6: All known nitrite-oxidizing Nitrospirales belong to the genus Nitrospira. In contrast, only two OTUs identified in this study belong to this genus. Furthermore, this statement as presented right now seems to have little relation to the discussion of Thermomicrobiaceae presence, which as stated correctly are sulfate reducers. Also, they do not really "remove" nitrogen, but convert it to nitrate, which in aerobic systems will accumulate.
- lines 291-4: Why is this change in the oxygenation regime never discussed in the increase (or up-migration) in anammox activities?

- Figure 2: The last sentence of the legend fails to mention "at 5 cm".
- Figure 4: Why are Planctomycetes not included here as the whole MS revolves around anammox?
- Figure 5: according to this figure all Planctomycetes decrease in abundance, but according to the main finding of this study Brocadia abundances (at least of one OTU) strongly increase. Shouldn't this then also be visible from this figure?
- Figure S1 is not referenced anywhere in the text.

- Table S1: Please remove the k_, p_, c_, o_, f_, g_ from the table cells to make it more readable. The classification level is given in the table header, which should be repeated on each page. Why are the OTUs with decreased abundance sorted largest to smallest change, while the ones with increased abundance are sorted smallest to largest?

·

Basic reporting

Black et al. studied the effect of mussels on the microbial community in the sediment of the upper Mississippi river. Both targeted qPCR and non-targeted amplicon sequencing was used. It is not completely clear to me why the targeted approach was used first. The results are mostly shown inpercentages or fold-changes. Unfortunately, the authors do not write anywhere in the paper the total number of 16S rRNA reads used for these calculations (although there is some information in Figure S1). This makes it difficult to judge how relevant these changes are.

Line 31: “Denitrifying microorganisms showed no significant changes associated with mussel presence”. It is difficult to understand where this conclusion comes from, how were changes in 16S rRNA abundance coupled to denitrifying organisms? For some microorganisms it might be clear that they are able to denitrify but for others maybe not based on their partial 16S rRNA sequence.

Line 249: “AOA … typically dominate sediment niches with high NH4+ and C (Shen et al. 2014). In this review paper is indeed stated that this is the case, however only at acidic pH. I do not know the pH of the sediment used in this paper. In general, I believe it is often stated that a higher concentration of NH4+ favours growth of AOB (see also for example Martens-Habbena et al, 2009; Verhamme et al., 2011) which is in agreement with the described results.

Line 269: Black et al mention that they indeed find results that are in contrast to other studies. Unfortunately they do not discuss why this could be. Is it maybe something specific for the UMR?

Line 285: “Nitrospirales …… serve an important N-removal mechanism in high-N environments”. This suggests that Nitrospira are adapted to high nitrite concentrations, I think this is not the case. Nitrospira are well adapted to low substrate concentrations (Nowka et al., 2015; Schramm et al., 1999). But here are not always nitrite oxidizers meant so I wonder why this comment is here as an explanation for the observed results. Please clarify.

Experimental design

Line 68: Why would mussels only alter the quantity of anammox bacteria and not all nitrogen cycle microorganisms? And how can mussels influence the nitrite concentration in the sediment? As far as I know, these organisms cannot oxidize ammonia into nitrite.

Line 87: Samples were taken from two different locations. It is probably difficult to determine but is the sediment of these locations similar in abiotic parameters (i.e. pH, trace elements, depth, temperature).

Line 94: “sampling depth for batch 2 was selected based on the anammox abundance in batch 1”. Was only anammox abundance checked in this sample? It is not clear to me why the presence of anammox was taken as a parameter for the sampling of Batch 2.

Line 127: the primers used for the amplification of the bacterial 16S rRNA gene are mentioned here. Which primers are used for the archaeal gene?

Validity of the findings

Line 194: In this statistic assay 77288 OTUs were tested. Does this mean that all tests were performed with this number of OTUs? But how much is a 4 fold change in absolute numbers of OTUs?

Line 204: what is exactly meant with a “much smaller initial abundance”?

Line 230: You saw that one species of Brocadia became more abundant in the presence of mussels and another less. What do you exactly mean with the comment: “this highlight the importance of your targeted approach”? In your qPCR you target “all” anammox species.

Line 301: “mussel presence suppressed ammonia oxidizing archaea”; from which data is this conclusion drawn?

Comments to Figures and tables:
Table 1: Why are the Nitrospira OTUs in a table in the main text? And are all columns of this table really needed to show us the results discussed in the main text? Several OTUs do not even have a last column which makes this table difficult to read.

Figure 1: does the color of the different arrows mean something?

Figure 2: depths below 9 cm were not discussed at all in this paper and since there is only 1 sample used for this analysis, I would remove it from the figure.

Figure 4: In order to be able to put these data into perspective, it would be nice if the total number of analysed sequences would be given.

Figure 4: Was there no difference found in the abundance of Planctomycetes in the non-targeted 16S rRNA approach? Since this is one of the main targets for this study and since the depth of which the samples were taken were based on qPCR results targeting anammox bacteria and since this was one of the results in the targeted approach?

Figure 4 and 5: It took me quite some time to understand what figure 4 and 5 show. I would make this more clear in the text.

Line 520: “increase in Nitrospirae is the same when summing across all detected OUT’s”; I do not understand this sentence.


Minor comments:
Line 35: “these findings further our understanding….”. I think there is a word missing here.

Line 69: “influence the mass of available NH4+. Do you mean here the amount of ammonium?

Additional comments

In my opinion, the title is not really fitting to the main conclusions/goal of this study. In addition, a relatively large part of the introduction is spent on anammox but not on other N-cycle microorganisms. In view of the results shown and discussed in the paper, the introduction could be a bit less focussed on anammox.

---

## Round 0.2 · Minor Revisions

Your manuscript still has to be improved according to the indications provided by the reviewers. Special attention should be taken with the Materials and Methods section.

·

Basic reporting

This is a very well written manuscript, that might need some carefull revision over the Methods section, which appears not quite as well revised as the rest of the manuscript. Overall it reads very well, especially across the introduction and discussion sections.

Experimental design

no additional comments since first version

Validity of the findings

In general the reporting of the data appears valid and sound, besides the wrong functional classification on Crenothrix (see below).

Additional comments

- Complete ammonia oxidizers (comammox) belong to the genus Nitrospira (more specifically sublineage 2), not Crenothrix! Crenothrix is a gammaproteobacterial methane oxidizer, and the classification of its amoA/pmoA in the comammox amoA group was wrong (See publications by van Kessel et al. and Daims et al., 2015). As you detected Crenothrix by 16S rRNA sequencing only and don't have amoA sequencing data (which is the only reliable method for comammox right now) you will need to change all instances where you discuss appearance of comammox, who's potential presence you can discuss, but not conclude from your data.

- L98: Many AOB can also use urea, as can comammox Nitospira, and even some canonical Nitrospira, which then provide the ammonia formed to AOB in their vicinity (see Koch et al., 2015)
- L106: As much as I like to be cited, this is a genome paper. Better use our recent review (Daims et al. 2016) or Daims et al. 2001.
- L113: revise sentence, strange connection "..., and..." (missing "this"?)
- L115: This is not true, as Nitrospina have the same type of NXR, which is even more similar to the anammox enzyme (Lücker et al. 2012). Furthermore, the NXR type might indicate the nitrite affinity, but the oxygen affinity (which is required as terminal electron acceptor for nitrite oxidation) depends on the type of terminal oxidase.
- L122: Please remove all instances of Crenothrix here. Only the amoA type that was found in the Nitrospira genome had been assumed to belong to Crenothrix. Crenothrix still is a methane oxidiser, but uses a canonical pmoA and is not a Nitrospira-like organism, but a Gammaproteobacterium.
- L130: N-damo archaea were discovered by Raghoebarsing et al. 2006 and Haroon et al. 2013.
- L165 and 167: please change "16S" to "16S rRNA gene", also at all instances later in the manuscript.
- L168: I guess you quantified 16S rRNA gene copy numbers. Now it reads as if you quantified RNA.
- L174: This is unclear to me. Did you use single, isolated plasmids from your clone library, or isolated mixed (plasmid) DNA from all clones without singularisation and checking for the correct insert? Please clarify this. You partly go into more detail below, so I would advise restructuring this paragraph. Even in the next paragraph it remains unclear if you used single or mixed plasmids.
- L176: The clone library (or isolated plasmid) can not serve as standard curve, it can be used to generate one.
- L231: ...comammox genus Crenothrix...
- L341: Crenothrix is a methane oxidiser and not involved in N-cycling.
- L446: How can OMZs have a sediment surface?
- L478: replace "which" by "who".
- L480-2: I get what you mean, but this sentence is unclear. Different Nitrospira can only adapt to different nitrite and oxygen concentrations.
- L483: As you find so many different OTUs within the Nitrospiraceae which are not affiliated with the genus Nitrospira, I would advise to indicate more clearly that the OTUs you indicate here really belong to the genus.
- L508: As I guess you really detected Crenothrix based on 16S rRNA sequences I am afraid you have to rewrite all the instances you have on comammox, as Crenothrix really is a methane oxidizer and does not belong to the genus (or even phylum) Nitrospira.
- L513: From the data presented here I wouldn't conclude that the role of anammox necessarily is small. As you dont have process rates I would not try to quantify them. Even low-abundance organisms mit be catalytically highly active.
- L515: I guess in this paragraph you could include your data on Crenothrix, as it could outcompete N-damo for methane in oxygenated sediments.
- L531: Please indicate how you classified conventional denitrifiers - I guess this is difficult based on 16S rRNA only.
- L535: This greater abundance of Nitrospira includes Crenothrix, I guess, as according to the data presented Nitrospira increases slightly less then anammox.

·

Basic reporting

The most related AmoA gene of comammox in public databases used to be annotated as crenothrix pmoA. However, this does not mean that crenothrix is comammox Nitrospira. So far, we cannot conclude from a 16S rRNA sequence of Nitrospira if it can oxidize ammonia completely to nitrate of if it is a real nitrite oxidizer. This has to be changed in the main text, because now the conclusion of the presence of comammox is drawn from data where you cannot draw these conclusions from. The same applies for table 1, Crenothrix cannot be classified as comammox.

Line 117: compatitve = competitive
Line 120: I would rephrase this sentence; an anammox enrichment culture would not be an anammox enrichment culture if anammox microorganisms would not be there
Line 135: Rephrase this sentence please.
Line 443: I think you mean here that the increase in number of Brocadia C-deposition by the mussels?
Line 514: I do not understand how this study is showing that anammox plays a small but significant role in N-transformations.

Experimental design

Was there in anyway checked if DNA quality and PCRs were similar in all samples? Do you think that no biases were introduced by the way the metagenomes were obtained from samples with and without mussels (no additional handling to remove mussels, difficulties with sampling etc)? I do see that there is no significant difference in the number of sequences obtained from the samples.

Validity of the findings

The comammox bacteria published by van Kessel et al (2015) and Daims et al (2015) were classified as Nitrospira and definitely not classified as Crenothrix. The closest known amoA in public databases (by that time) was classified as pmoA, which definitely does not mean that Crenothrix is comammox. Based on 16S rRNA phylogeny, you find the comammox microorganisms known so far within the Nitrospira. In addition, comammox Nitrospira can also oxidize ammonia at ambient oxygen as shown in both papers.

Additional comments

Black et al. have improved their manuscript a lot compared to the first version. The additional results and discussion on different N-cycle microorganisms are major improvements. These additional results also clarify why some conclusions were drawn.

---

## Round 0.3 · Minor Revisions

As per the two reviewers, your paper needs some remaining minor revisions before being accepted for publication.

·

Basic reporting

no changes necessary after revision

Experimental design

no additional comments since last version

Validity of the findings

There are a few last issues in the interpretation of the data (see below), but in general I do conform with the authors interpretation and representation of the data.

Additional comments

- L32: Change to "archaeal" family
- L89: This is a quite bad introduction of nitrification, as it reads as if ammonia oxidation is the only step involved. Nitrite-oxidising bacteria are not mentioned until line 100, and here it is not even indicated that this is the second step of nitrification, but only states that they also "contribute to N-transformations".
- L106: You already introduce the ureolytic activity of some NOB here, which you use later in the discussion, but not complete ammonia oxidation by comammox Nitrospira. Why not? It would fit well here and would make the discussion later easier to read. You can still come bac to the comammox-anammox coculture later (line 124), without much repetitiveness.
- L115: The interpretation of the similar NXR of Nitrospira and anammox as presented here does not make any sense. ALL Nitrospira have this type of NXR (see Koch 2015, Daims 2015, van Kessel 2015), not only N. defluvii!
- L117-9: While all facts cited here are correct, the interpretation as presented now is wrong. If Nitrospira reduces nitrate under anoxic conditions, it FORMS nitrite and thus does not compete, but cooperate with anammox, as they benefit from this additional nitrite supply.
- L163 and 166: "Sediment was removed" this sounds as if sediment was separated from the microbial biomass prior to DNA extraction, and not sampled for it.
- L323-4: This part reads a bit confusing, as for all other OTUs the increase from without to with mussels is mentioned, and for this Brocadia it is suddenly the other way around.
- L380-3: Please revise this sentence. It basically states that mussels are responsible for the mussel-associated microbiome, which seems quite obvious.
- L385: I find this a funny use of "equitability"? What exactly is meant here? Evenness?
- L404-5: "has BEEN shown"
- L431: Change "is" for "are"
- L434: Change "in" to "of" or "belonging to/affiliated with"
- L455: Please revise sentence, appears to be remnant of too much rephrasing.
- L457: Misplaced comma after likely.
- L476: You maybe should move the citations to the end of the sentence. My brain always ends up reading "are inhibited by ammonia concentrations" only.
- L483: What does Nitrososphaera adapt to here? Low nitrite? Furthermore, a citation for this organism is missing here.
- L502: This conclusion as drawn/discussed here is wrong, as the conversion of urea to ammonia by NOB, with subsequent oxidation to nitrite by AOB works only if this nitrite then can be used by Nitrospira as energy source. Thus, urea is an indirect energy source for NOB, but anammox and Nitrospira still compete for the nitrite formed. The only difference is that there is an additional ammonia/nitrite source available.
- L512: As you are referring to our study here (van Kessel et al., 2015), please note that ALL Nitrospira in this bioreactor had the genomic potential for ammonia oxidation, but we of course had no way to show if they all were growing by ammonia oxidation.
- L516: You are referring to nitrate-dependent DAMO here, please clarify.
- L533: Please include a short discussion/speculation on the reason for this lack of denitrification here. Not enough organic carbon substrates? Limited nitrate formation due to anammox and lack of nitrite oxidation, e.g. due to limited oxygen availability?
- L562: exchange "is" for "are"
- L571: Cascade, not cascades.

·

Basic reporting

The manuscript improved again and reviewer comments were taken into account. I therefore have only minor remarks on this version of the manuscript!

L29: “ammonium oxidizing bacteria and archaea family Nitrosomonadaceae and genus Candidatus Nitrososphaera”. Please rephrase this sentence; I read now that Nitrosomonadaceae are archea.

L68: Ammonia and ammonium are always in equilibrium; so both increase when mussels excrete ammonia.

L115: please change into “this is especially true….”.

L119: “suggesting Nitrospira ….. variety of niches”. I do not really see why this is proven by the first part of the sentence.

L186: I assume that you tested with melting curves if the products obtained in the qPCR were really the right products. This is not indicated in the M&M, I think it is good to add this in the M&M if you did.

L268: than in no mussel samples.

L284: please remove so-called.

L289: remove “in figure 3A and 3B”.

L319: This sentence has to be rephrased; I do not understand this sentence anymore. What is meant with “increased from a low average count of…?

L323: Is there any more information about these two Brocadia species? One is increasing with and the other without mussels? But all together, Brocadia abundance increases with mussels being present. The way it is written here is a bit unclear.

L343: I think you mean greater abundance here?

L509: I would not dare to say that it is likely that some of the Nitrospira species are actually comammox Nitrospira just because anammox is also present. I think there is really more information needed. The study you refer to is a completely different system, so I would not automatically assume that there is the same link.

L523: Damo organisms decrease in abundance in sediments with mussels. Is this only caused by the increase in anammox?

Experimental design

no comment

Validity of the findings

no comment

---

## Round 0.4 · accepted · Accept

Thank you for improving your manuscript and congratulations for your work.